# Evaluation of hyper-attenuated S19 poliovirus strains for use in poliovirus neutralization assays

**Nicholas Wiese**[1,2], **William Hendley**[1], **Basit Jafri**[1], **Kathryn A.V. Jones**[1], **Giovanna Sifontes**[1], **Sandra Valdez**[1,3], **Yiting Zhang**[1], **Bernardo A. Mainou**[1]*

1 Polio and Picornavirus Branch, Division of Viral Diseases, Centers for Disease Control and Prevention, Atlanta, Georgia, United States of America, 2 United States Public Health Service, Rockville, Maryland, United States of America, 3 Tanaq Contracting Agency to the Division of Viral Diseases, Atlanta, Georgia, United States of America

* bmainou@cdc.gov

## Abstract

Poliovirus eradication efforts have relied on effective vaccination campaigns and stringent laboratory containment of poliovirus strains. Serological testing is the primary means to address population immunity, vaccine efficacy, and individual poliovirus exposures. However, strict containment requirements for Sabin type 2 and wild poliovirus types 2 & 3 limit access to serological testing. To address these limitations, we evaluated the performance of hyper-attenuated, temperature-sensitive S19 poliovirus strains (S19-PV), which retain the antigenic characteristics of Sabin and Salk strains but can be handled under biosafety level 2 (BSL-2) conditions. Using a modified microneutralization assay, we performed a comparative validation of S19-PVs to their Sabin and wild-type counterparts using WHO international standards, in-house control reagents, and human sera. Neutralizing antibody titers showed strong concordance between S19-PV and reference strains (R²=0.93–0.99), with high intra- and inter-assay reproducibility (CVs<10%). Our findings demonstrate that S19-PV strains are antigenically equivalent to Sabin and wild-type viruses and suitable for use in neutralization assays to determine levels of poliovirus neutralizing antibodies outside of strict containment laboratories. Adoption of these strains can significantly expand global capacity for poliovirus serological testing, supporting surveillance and immunization activities in the final stages of poliovirus eradication.

## Introduction

Poliovirus, a non-enveloped, positive-sense, single-stranded RNA virus in the *Picornaviridae* family, replicates in the oropharynx and gastrointestinal tract and rarely infects the central nervous system [1]. Poliomyelitis, caused by three antigenically distinct poliovirus serotypes, has clinical manifestations which include fever, sore throat, headache, vomiting, and limb pain [2,3]. In unvaccinated people, less than 1%

**Data availability statement:** All relevant data are within the paper and its Supporting Information files.

**Funding:** The author(s) received no specific funding for this work.

**Competing interests:** The authors have declared that no competing interests exist.

of poliovirus infections result in acute flaccid paralysis, an often-permanent paralytic illness affecting the lower limbs [4]. Owing to the potentially severe clinical outcomes and that humans are the only natural reservoir, vaccination emerged as the central strategy in global eradication efforts [5].

There are two types of poliovirus vaccines, inactivated poliovirus vaccines (IPV) and oral poliovirus vaccines (OPV), both of which can provide lifelong immunity against poliomyelitis [6,7]. While remarkable progress has been achieved, including eradication of wild poliovirus type 2 in 2015 [8,9] and wild poliovirus type 3 in 2019 [10], wild poliovirus type 1 remains endemic in Afghanistan and Pakistan [11]. Eradication efforts are also hampered by vaccine derived polioviruses (VDPVs), which arise when OPVs regain neurovirulence through genetic changes when administered to immunocompromised individuals or in areas where multiple transmission cycles can occur due to poor water sanitation or hygiene practices as well as areas of low immunization coverage [12,13].

To prevent the accidental release of poliovirus from laboratory facilities, strict containment measures have been outlined by the United States National Authority for Containment of Poliovirus (NAC) [14,15]. Currently, laboratory work in the United States using wild-type (Salk) serotype 2 and 3 polioviruses, as well as Sabin type 2 polioviruses, must be conducted under strict containment conditions (e.g., biosafety level 3, BSL-3) to comply with guidelines set by the NAC. Laboratory work using serotype 1 polioviruses including Salk and Sabin strains, and Sabin serotype 3 poliovirus can be conducted under biosafety level 2 (BSL-2) laboratory conditions with reduced containment measures. Laboratory work with novel oral poliovirus vaccine (nOPV) strains is exempt from strict containment and can be performed under BSL-2 laboratory conditions.

To address containment limitations associated with poliovirus research, the Medicines and Healthcare Products Regulatory Agency (MHRA, UK) engineered hyper-attenuated, temperature-sensitive poliovirus strains, known as S19-PV [16]. These strains possess structural proteins that match their corresponding Salk and Sabin poliovirus strains [17], are temperature sensitive, replicate poorly in non-human primates, and maintain the antigenic properties of their corresponding Sabin and wild-type poliovirus counterparts [16]. Like nOPVs, S19-PV strains are exempt from strict containment requirements, allowing laboratory work with these strains under BSL-2 conditions [18].

The development of the S19-PV strains could expand access to laboratory work with poliovirus to facilities that do not have access to a BSL-3 laboratory. Poliovirus serology testing, which assesses poliovirus immunity through detection of serotype-specific neutralizing antibodies [19,20], requires a BSL-3 laboratory to ascertain levels of type 2 poliovirus neutralizing antibodies. Serology plays a vital role in confirming seroconversion following vaccination, in seroprevalence surveys to determine gaps in poliovirus population immunity, assessing immunity to poliovirus in individuals, and in rare instances in diagnosing individual poliovirus exposure [21–24]. Poliovirus serology is also used to determine the efficacy of new vaccines, vaccine adjuvants, antivirals, or changes in vaccination schedules in inducing robust

poliovirus immunity [25–27]. Standard poliovirus serology relies on neutralization assays using either wild-type or Sabin poliovirus strains [28]. Due to the strict containment required to perform laboratory work with type 2 polioviruses, as well as wild-type 3 polioviruses, few laboratories are equipped to perform comprehensive poliovirus serology testing against all 3 poliovirus serotypes. In the United States, no commercial laboratories provide poliovirus serology testing against all poliovirus serotypes. As such, there is a need for alternatives that allow poliovirus serology testing outside strict biological containment environments [16,17,29].

Here we present data characterizing S19-PVs for use with a poliovirus microneutralization assay to assess levels of serotype-specific neutralizing antibodies. We show that S19-PV strains with both Sabin and wild-type (Salk strains) structural proteins have similar antigenic profiles to reference Sabin and wild-type polioviruses when tested with human, animal, and control sera. Intra-assay and inter-assay precision assessments of the S19-PVs show that they are a robust tool for use with poliovirus microneutralization assays. Together, our data show that the performance characteristics of the S19-PVs are comparable to Sabin and wild-type polioviruses, making them a useful tool to assess levels of poliovirus neutralizing antibodies outside of strict laboratory containment.

## Materials and methods

### Cell and virus propagation

HEp-2C cells were cultured in Modified Eagle's Medium (MEM, Gibco) supplemented with 2% fetal bovine serum (FBS, R&D Systems), 2 mM L-glutamine (Gibco), 10 mM HEPES Buffer (Gibco), 100 units/mL penicillin (Gibco), and 100 µg/mL streptomycin (Gibco). Cells were maintained at 35°C in a humidified 5% $CO_2$ environment.

S19 poliovirus strains S19-S1 (PP_068131), S19-S2 (PP_068132), S19-S3 (PP_068133), S19-Mahoney (PP_068134), S19-MEF1 (PP_068135), and S19-Saukett (PP_068136) were provided by the Medicines and Healthcare Products Regulatory Agency (MHRA, formerly the National Institute of Biological Standards and Control, UK) [16,17]. Individual virus strains were used to infect 75 cm$^2$ tissue-culture flasks containing confluent HEp-2C cells and incubated at 33°C with 5% $CO_2$ until cytopathic effect (CPE) was observed. Cells underwent three freeze-thaw cycles, supernatants were clarified by centrifugation at 4°C for 20 min at 1,500 × g. Virus stocks were aliquoted and stored at −80°C for long-term storage.

Oral polio vaccine Sabin type 1 (Catalog # 01/528 & 10/164) (AY_184219), type 2 (Catalog # 01/530 & 15/296) (AY_184220), and type 3 (Catalog # 01/532 & 10/168) (AY_184221) were obtained from MHRA. Wild poliovirus type 1 (Mahoney, AY_082689), 2 (MEF-1, AY_082677), and 3 (Saukett, L_23848) are part of the reference strain collection in the Polio and Picornavirus Branch at the Centers for Disease Control and Prevention (CDC). Sabin and wild-type polioviruses were amplified in HEp-2C cells at 35°C as described previously [28]. Virus stock titers were calculated using the Spearman-Kärber method [30].

### Reagents

The World Health Organization (WHO) International Standard containing human neutralizing antibodies to poliovirus types 1–3 (anti-PV, 82/585), PV-positive rat serum from a historical IPV immunization test (Rat positive), negative rat sera (Rat Negative), and WHO International Standard anti-EV71 serum (14/138, 14/140, and 13/238) were obtained from MHRA. Standards 14/138, 14/140, and 13/238 were prepared as previously described [31], have assigned unitage of 1,000, 1,090, and 300 International Units (IU) of anti-EV71 antibodies per ampoule and will be referred to as anti-EV71 high pool one, anti-EV71 high pool two, and anti-EV71 low, respectively.

A pool of serum samples (in-house reference sera) with known levels of poliovirus neutralizing antibodies were used as positive control samples [28]. Residual human serum samples from historical poliovirus vaccination studies were used as test samples for validation of S19-PV strains. These samples were accessed between 11/15/2022–02/08/2024 for research purposes. The authors had no access to information that could identify individual participants during or after data

collection. This activity was reviewed and approved by the Centers for Disease Control and Prevention (CDC) Human Subjects Research Protection Office (HRPO) and study was reviewed and approved by the Seoul National University Hospital Institutional Review Board IRB H-2305-046-1430 (See 45 C.F.R. part 46.114; 21 C.F.R. part 56.114).

### Poliovirus microneutralization assay

The poliovirus microneutralization assay was performed as previously described when using Sabin or wild-type poliovirus strains [28]. For serological testing using S19-PV strains, the microneutralization assay was modified by incubating cells at 33°C instead of 35°C and extending the incubation time from 5 to 7 days. Linearity of the control reagents were assessed by performing sixteen two-fold serial dilutions, starting at 1:2 and ending with a final dilution of 1:65,536 for end-point titer analysis. Human sera samples used for direct comparison were assessed in the same manner but with a starting dilution of 1:8 and ending with a final dilution of 1:262,144.

### Data analysis

All raw data are provided in supplemental files. Data and statistical analysis including correlation coefficients, coefficients of variance, and standard deviations were calculated using GraphPad Prism Software (v10).

### Ethics statement

This activity was reviewed and approved by the Centers for Disease Control and Prevention (CDC) Human Subjects Research Protection Office (HRPO) and study was reviewed and approved by the Seoul National University Hospital Institutional Review Board IRB H-2305-046-1430 (See 45 C.F.R. part 46.114; 21 C.F.R. part 56.114). An approved Informed Consent Form was used and the informed consent process was conducted based on sufficient explanation under no coercion or unfair influence and allowing sufficient opportunity for study participation.

## Results

### Analysis of control reagents

To perform an initial comparison of the S19-PVs to reference Sabin and wild-type (Salk) polioviruses, WHO reference reagents provided by MHRA were tested against reference Sabin strains (types 1, 2, and 3), reference wild-type strains (Mahoney, MEF-1, Saukett), and the S19-PVs with Sabin (S19-S1, S19-S2, S19-S3) or wild-type (S19-W1, S19-W2, S19-W3) structural proteins to define the antigenic profile of the different poliovirus strains using a microneutralization assay in HEp-2C cells.

Anti-PV serum was seropositive against all Sabin viruses tested, with mean neutralizing antibody titers ($log_2$) of 12.72, 11.17, and 9.50 against S19-S1, S19-S2, or S19-S3, and 10.83, 10.83, and 9.50 against Sabin 1, 2, or 3, respectively (Fig 1, closed squares). Anti-EV71 high pool 1 sera was seronegative against all Sabin viruses tested, with mean neutralizing antibody titers of 1.67, 1.33, and 2.00 against S19-1, S19-2, or S19-3, and neutralizing antibody titers 1.83, 1.83, and 1.83 against Sabin 1, 2, or 3, respectively (Fig 1, open triangles). Interestingly, Anti-EV71 Low sera was seropositive against all Sabin viruses tested, with mean neutralizing antibody titers of 7.50, 8.33, and 5.17 against S19-S1, S19-S2, or S19-S3 and neutralizing antibody titers 7.17, 8.17, and 5.00 against Sabin 1, 2, or 3, respectively (Fig 1, closed triangles).

Rat positive sera was seropositive against all Sabin viruses tested, although antibody titers were higher against type 2 than types 1 and 3 with mean neutralizing antibody titers ($log_2$) of 3.33, 9.33, and 3.83 against S19-S1, S19-S2, or S19-S3 and neutralizing antibody titers ($log_2$) 3.17, 9.83, or 3.17 against Sabin 1, 2, or 3, respectively (Fig 1, closed circles). Rat negative sera was seronegative against all Sabin viruses tested (Fig 1, open circles). Comparison of neutralizing antibody titers between the S19 strains and their Sabin counterparts (e.g., S19-S1 to Sabin 1, S19-S2 to Sabin 2, and S19-S3 to Sabin 3), showed a strong correlation between neutralizing antibody titers of the S19 and Sabin Strains ($R^2$-values of the best fit lines were 0.99, 0.99, and 0.96 for types 1, 2, and 3, respectively).

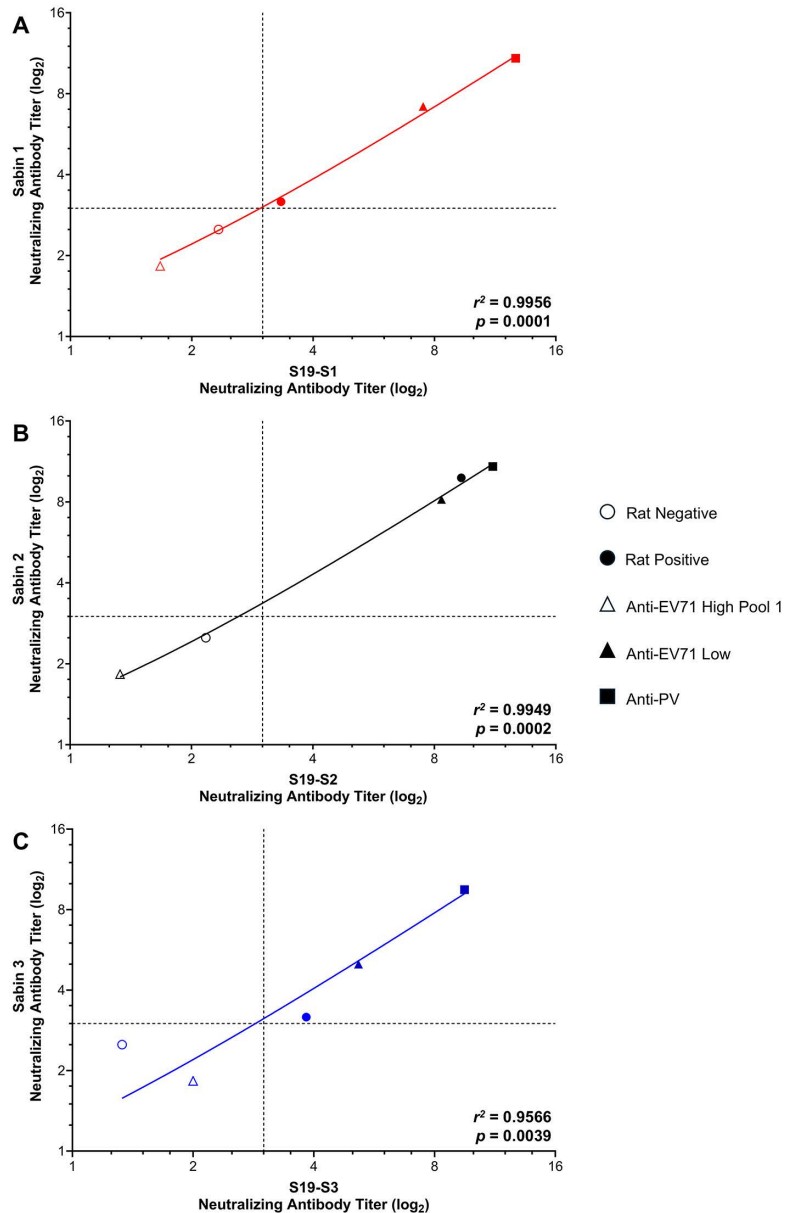

**Fig 1. Neutralizing antibody titers of control sera against Sabin polioviruses.** Control sera (PV-negative rat serum (open circle), PV-positive rat serum (closed circle), anti-EV71 high pool 1 (open triangle), anti-EV71 low (closed triangle), and anti-PV (closed square)) were assessed for levels of poliovirus neutralizing antibody titers against Sabin and S19-Sabin type 1 **(A)**, type 2 **(B)**, and type 3 **(C)**. Data are shown as the average neutralizing antibody titer ($\log_2$) for samples tested in triplicate for three independent experiments (S19-PV strains) and in triplicate for one independent experiment (Sabin strains). Dashed lines indicate the lower limit of detection for the assay. $R^2$ and p-values are shown for each serotype.

Anti-PV serum was seropositive against all wild-type viruses tested, with mean neutralizing titers ($\log_2$) of 11.39,12.39, and 9.39 against S19-W1, S19-W2, and S19-W3, and 10.50, 12.83, and 10.50 against Mahoney, MEF-1, or Saukett, respectively (Fig 2, closed square). Anti-EV71 high pool 2 sera (14/140) was seropositive against all viruses tested, with mean neutralizing antibody titers ($\log_2$) of 7.28, 9.33, and 5.50 against S19-W1, S19-W2, or S19-W3 and neutralizing antibody titers ($\log_2$) of 7.50, 8.83, and 5.50 against Mahoney, MEF-1, or Saukett, respectively (Fig 2, open triangle).

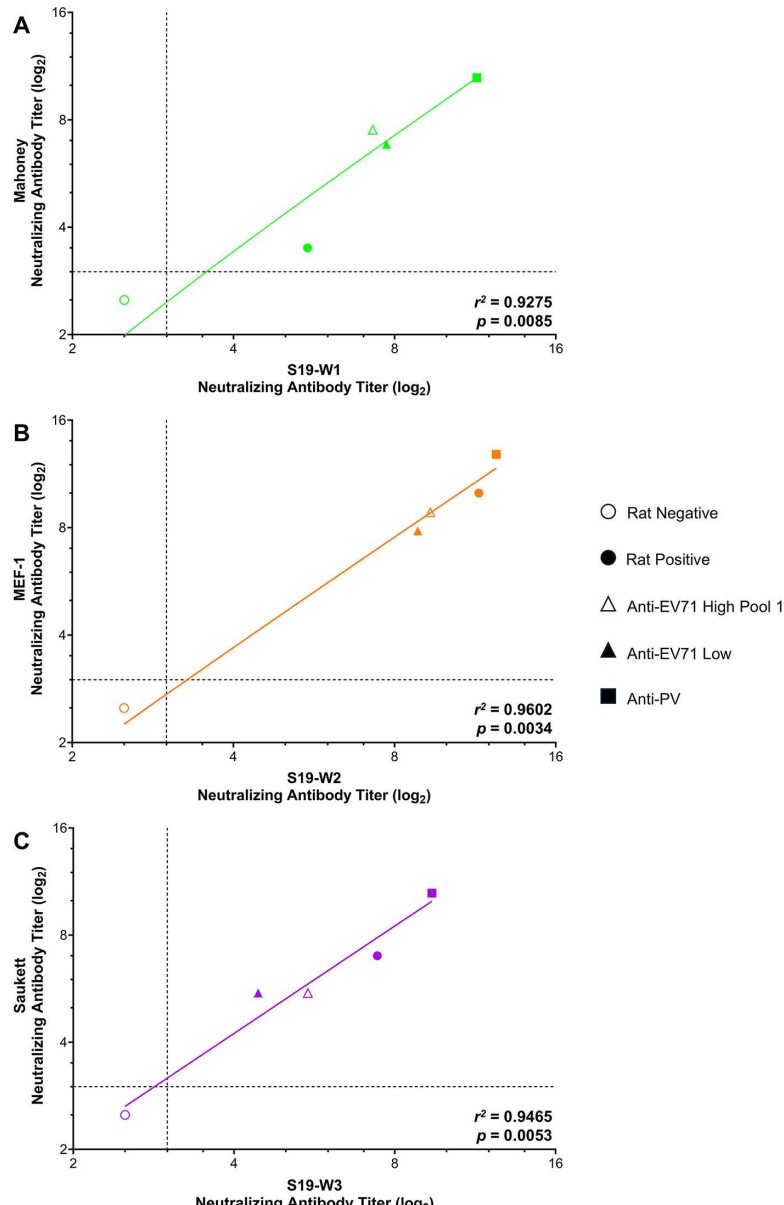

**Fig 2. Neutralizing antibody titers of control sera against Salk polioviruses.** Control sera (PV-negative rat serum (open circle), PV-positive rat serum (closed circle), anti-EV71 high pool 1 (open triangle), anti-EV71 low (closed triangle), and anti-PV (closed square)) were assessed for levels of poliovirus neutralizing antibody titers against Salk and S19-Salk Mahoney **(A)**, MEF-1 **(B)**, and Saukett **(C)**. Data are shown as the average neutralizing antibody titer ($log_2$) for samples tested in triplicate for three independent experiments (S19-PV strains) and in triplicate for one independent experiment (Salk strains). Dashed lines indicate the lower limit of detection for the assay. $R^2$ and p-values are shown for each serotype.

Neutralizing antibody titers with anti-EV71 pool 2 are in contrast with those observed against anti-EV71 pool 1 with Sabin strains. The anti-EV71 low standard (13/238) was seropositive against all viruses tested with mean neutralizing antibody titers ($log_2$) of 7.72, 8.84, and 4.44 against S19-W1, S19-W2, or S19-W3 and 6.83, 7.83, and 5.50 against Mahoney, MEF-1, or Saukett, respectively ([Fig 2](), closed triangle).

The rat positive sera was also seropositive against all wild-type viruses tested, and similar to that of the Sabin results, had higher antibody titers against type 2 than types 1 or 3 with mean neutralizing titers ($\log_2$) of 5.50, 11.50, and 7.42 against S19-W1, S19-W2, or S19-W3, and 3.50, 10.00, and 7.00 against Mahoney, MEF-1, or Saukett, respectively (Fig 2, closed circle). The PV-negative rat serum was negative in all instances (Fig 2, open circle). Comparison of neutralizing antibody levels of S19-W1 to Mahoney, S19-W2 to MEF-1, and S19-W3 to Saukett, revealed a high correlation, with $r^2$-values of the best fit lines of 0.93, 0.96, and 0.95, respectively, indicating a high degree of concordance between calculated titers.

## Analysis of in-house reference serum

To assess intra-run reproducibility, human in-house reference sera were tested in three separate independent experiments against S19-Sabin strains (9 technical replicates) and S19-wild type strains (4–5 technical replicates) (Fig 3). Neutralizing antibody titers were calculated, and coefficients of variation (CV) were determined ((standard deviation of titers/mean titer) × 100%). CVs were 5.82, 8.79, and 9.52% for each S19-S1 run, 3.74, 6.96, and 7.84% for each S19-S2 run, and 10.32, 5.63, and 7.07% for each S19-S3 run. CVs were 4.75, 8.10, and 8.46% for each S19-W1 run, 7.19, 5.51, and 5.51% for each S19-W2 run, and 7.19, 7.54, and 5.51% for each S19-W3 run.

To assess inter-run reproducibility, the mean and standard deviation of each run for each serotype was calculated. For S19-Sabin, the CVs were 8.83 (S19-S1), 8.09 (S19-S2), and 5.93% (S19-S3). For S19-wild, the CVs were 5.44 (S19-W1), 5.44 (S19-W2), and 6.78% (S19-W3) (Fig 3). CVs for both intra- and inter-run reproducibility were below 20%, indicating high levels of reproducibility within and between runs for S19 Sabin and Salk strains. When assessing overall neutralizing antibody levels between Sabin and Salk strains, no significant differences were observed with any of the 3 poliovirus serotypes.

## Concordance between patient samples

To assess the performance of S19 strains against serum samples with various levels of neutralizing antibodies, we tested fifty human serum samples against Sabin strains, and forty-four human serum samples against wild type strains. Levels of neutralizing antibody titers for each sample were determined and compared between the Sabin and S19-Sabin strains (Fig 4) and wild-type and S19-wild type strains (Fig 5). Antibody titers highly correlated between all viruses tested, with correlation coefficient values of 0.970 (S19-S1 vs. Sabin), 0.968 (S19-S2 vs. Sabin 2), 0.985 (S19-S3 vs. Sabin 3), 0.939 (S19-W1 vs. Mahoney), 0.979 (S19-W2 vs. MEF-1), and 0.935 (S19-W3 vs. Saukett). These data indicate a high degree of concordance in the levels of neutralizing antibodies detected when using Sabin or S19-Sabin strains and wild-type and S19-wild type strains.

## Discussion

Containment of poliovirus and poliovirus infectious materials are an essential component of poliovirus eradication efforts. While one goal of poliovirus containment is to limit the number of facilities working with containable materials, certain laboratory work is required to inform poliovirus eradication efforts. These efforts include testing laboratory staff to assess their levels of neutralizing antibodies against all three poliovirus serotypes prior to working with poliovirus. The engineered S19-PV strains retain the antigenic properties of their Sabin and Salk counterparts, making them ideal candidates for serological testing for poliovirus neutralizing antibodies, vaccine production, and molecular detection controls outside of high biological containment.

Here, we assessed the antigenic profile of S19-PV strains in comparison to the Sabin and wild-type (Salk) strains used in current serological testing. The antigenic profile of all S19-PVs are similar to the antigenic profiles of their Sabin or wild-type counterparts. Control WHO International Standard reagents for anti-poliovirus types 1, 2, and 3 had similar

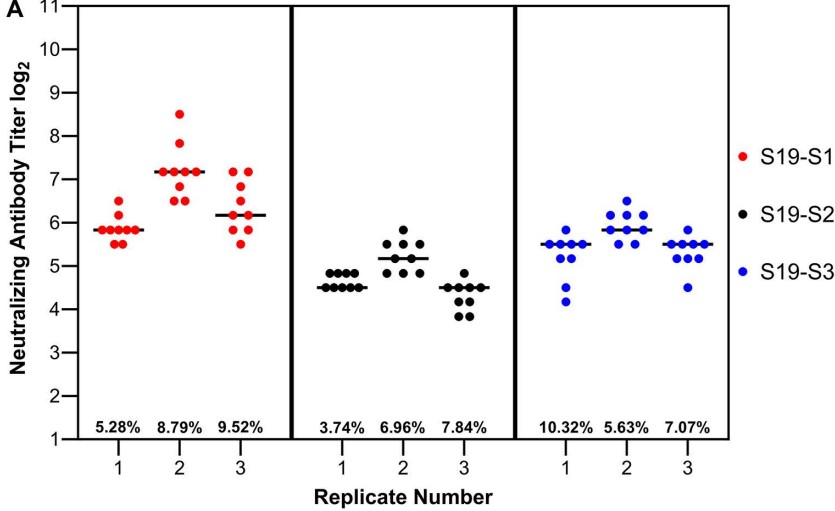

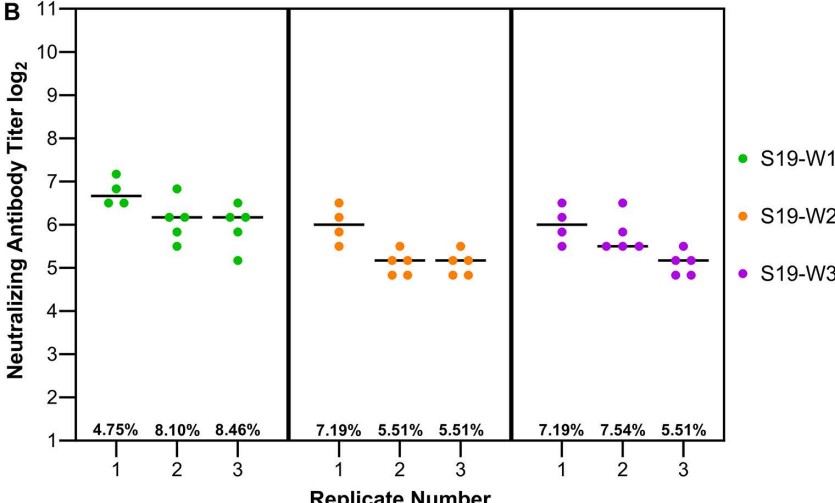

**Fig 3. Intra-run reproducibility of S19-PV adapted microneutralization assay.** Poliovirus neutralizing antibody titers were determined for human in-house reference sera against **(A)** S19-Sabin (types 1, 2, and 3) or **(B)** S19-Salk (Mahoney, MEF-1, and Saukett). Data are shown for individual neutralizing antibody titers ($log_2$) for each virus strain performed in triplicate for three independent experiments. Mean neutralizing antibody titers for each condition is indicated by a solid black line. Percent coefficient of variation (%CV) is shown for each replicate.

neutralizing titers between the S19-PV strains compared to Sabin or wild-type polioviruses. Negative control reagents were also seronegative when tested using S19-PVs, Sabin, and wild-type polioviruses. Lastly, the anti-EV71 WHO International Standard, which is reported to have neutralizing antibody titers against poliovirus (internal communication) had comparable neutralizing antibody titers between all viruses tested. We did note differences in neutralizing antibody levels with the anti-EV71 high pool standards. While high pool 1 (14/138) was seronegative against all Sabin-type viruses, high pool 2 (14/140) had significant neutralizing antibodies against all wild-type polioviruses. While these standards serve as reference reagents against EV-A71, the pools of sera used to create the standards were from 10 distinct human serum samples [31]. These results highlight the inherent variability of antibody composition in human sera, especially with

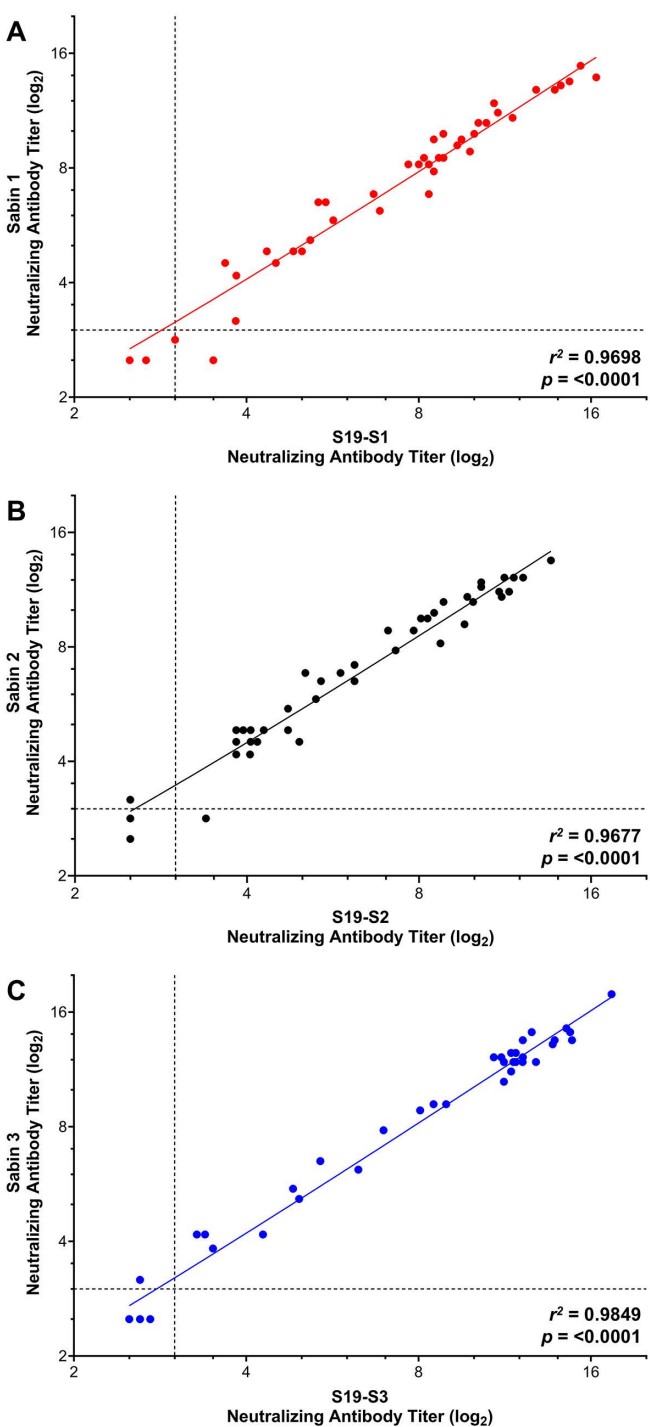

**Fig 4. Neutralizing antibody titers from human serum samples against Sabin polioviruses.** Human serum samples were assessed for the presence of poliovirus specific neutralizing antibody titers against Sabin and S19-Sabin type 1 **(A)**, type 2 **(B)**, and type 3 **(C)**. S19-PV strains were tested with three replicates within three independent experiments. Data are shown as the average neutralizing antibody titer (log$_2$) for samples tested in triplicate for three independent experiments (S19-PV strains) and in triplicate for one independent experiment (Sabin strains). Dashed lines indicate the lower limit of detection for the assay. $R^2$ and p-values are shown for each serotype.

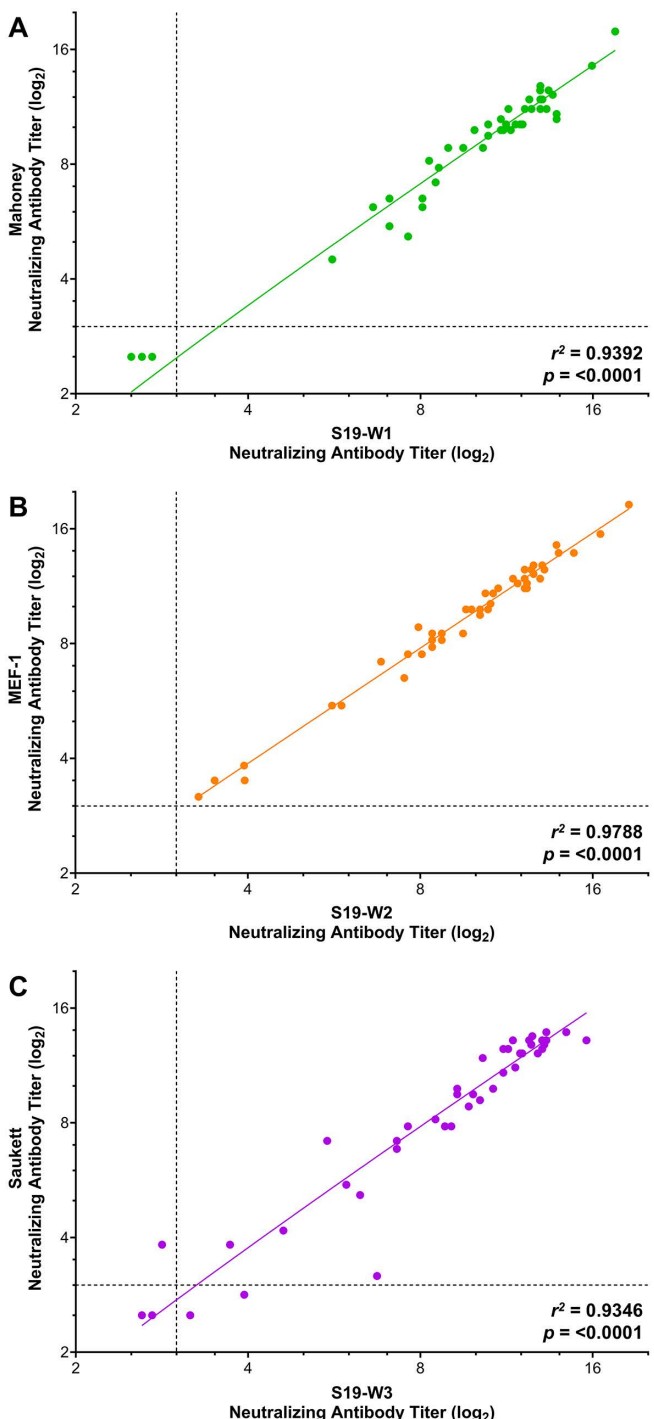

**Fig 5. Neutralizing antibody titers from human serum samples against Salk polioviruses.** Human serum samples were assessed for the presence of poliovirus specific neutralizing antibody titers against Salk and S19-Salk Mahoney **(A)**, MEF-1 **(B)**, and Saukett **(C)**. Data are shown as the average neutralizing antibody titer ($\log_2$) for samples tested in triplicate for three independent experiments (S19-PV strains) and in triplicate for one independent experiment (Salk strains). Dashed lines indicate the lower limit of detection for the assay. $R^2$ and p-values are shown for each serotype.

regards to highly prevalent human viruses. We show that there is no relevant effect on neutralizing antibody activity in the S19-adapted microneutralization assay given the decrease in incubation temperature from 35°C to 33°C compared to the Sabin or Salk microneutralization assay [32].

When testing human serum samples with a wide range of poliovirus neutralizing antibody titers, there was a high level of concordance between S19-PV and counterpart strains, with correlation coefficient values ranging from 0.93–0.98. There were, however, some discordant results in the human test samples. In the S19-Sabin validation runs, there was one sample in S1 (2%), two samples in S2 (4%), and one sample in S3 (2%) with discrepant seroprevalence results (titers ≥3.00 log2 being seropositive). In the S19-wild validation runs, there were zero samples in W1, zero samples in W2, and three samples in W3 (6.8%) with discrepant seroprevalence results. Out of the collective 7 discordant results, 4 were false-positive (negative with Salk or Sabin strains but positive with S19-PV), and 3 false negatives (positive with Sabin or Salk strains but negative with S19-PV). In all instances of discrepant results, the reported neutralizing titers (S19-S1: 3.50; S19-S2: 2.50, 3.39; S19-S3: 2.61; S19-W3: 2.83, 3.17, 3.94) were near the lower limit of the reportable range for the assay. These data suggest that while the S19-PV immunogenicity highly correlates with their Sabin or Salk counterparts, small variations at the lower limits of detection of the assay can be expected and could impact serological results.

There were several limitations associated with this study. First, the S19-PV strains have strict conditions for amplification and preparation. While reversion to a neurovirulent form has been reported to be very low [16], it is recommended by MHRA to limit amplification of S19-PV to 2–3 passages from the original stock. We suggest that if additional passages are required to amplify the S19-PV strains, verification of new stocks through sequence analysis is performed as described [17] as well as confirmation that antigenicity of new stocks remains unchanged. Second, when performing the validation, samples were tested in triplicate over three independent runs when using the S19-PV strains as challenge virus, compared to a single testing with triplicate replicates when using Sabin or wild-type polioviruses. We could not test additional replicates in our standard microneutralization assay because sample volume was not sufficient with all specimens. Testing with Sabin or wild-type poliovirus could provide higher resolution in the similarity between neutralizing antibodies with the different polioviruses. We were also limited by access to sufficient volume of control reagents for expanded testing. While the neutralizing antibody titer results for anti-EV71 high pool 1 were concordant between S19 and Sabin types, we were unable to determine if the standard is also seronegative against S19-wild and the reference wild-type strains. An additional limitation is the need for modifications to the standard microneutralization assay when using the S19-PV strains. Incubation length must be extended from 5 to 7 days to accommodate the lower replication kinetics of S19-PV. This can increase the likelihood of assay contamination and introduce possible variability between testing with S19-PV and Sabin or Salk strains.

In conclusion, we show the performance characteristics of the S19-PVs are comparable to those of Sabin and wild-type polioviruses and can be used as a critical reagent for assessing poliovirus neutralizing antibodies outside of high containment laboratory settings. Use of S19-PVs provide an increased level of safety for laboratory staff as well as the general public, while supporting increased global capacity for poliovirus serological testing in the final stages of poliovirus eradication.

## Supporting information

**S1 File. Supplemental Data for Fig 1. Data for control reagents and descriptive statistics corresponding to Sabin and S19-Sabin strains shown in Fig 1.**
(XLSX)

**S2 File. Supplemental Data for Fig 2. Data for control reagents and descriptive statistics corresponding to wild type and S19-wild type strains shown in Fig 2.**
(XLSX)

**S3 File. Supplemental Data for Fig 3. Data for intra assay variability and descriptive statistics for S19-PV strains shown in Fig 3.**
(XLSX)

**S4 File. Supplemental Data for Fig 4. Data for serum samples and descriptive statistics corresponding to Sabin and S19-Sabin strains shown in Fig 4.**
(XLSX)

**S5 File. Supplemental Data for Fig 5. Data for serum samples and descriptive statistics corresponding to wild type and S19-wild type strains shown in Fig 5.**
(XLSX)

## Acknowledgments

Authors would like to thank Javier Martin, Andrew McAdam, and others at the Medicines and Healthcare Products Regulatory Agency for generously providing the S19-PV strains and WHO control reagents. We thank Steve Oberste and Cara Burns for their leadership and support from the Polio and Picornavirus Branch and Division of Viral Diseases.

## Author contributions

**Conceptualization:** Nicholas Wiese, Bernardo A. Mainou.

**Data curation:** Nicholas Wiese.

**Formal analysis:** Nicholas Wiese.

**Investigation:** Nicholas Wiese, William Hendley, Basit Jafri, Kathryn A.V. Jones, Giovanna Sifontes, Sandra Valdez, Yiting Zhang, Bernardo A. Mainou.

**Methodology:** Nicholas Wiese, Bernardo A. Mainou.

**Project administration:** Nicholas Wiese.

**Resources:** Nicholas Wiese, William Hendley, Basit Jafri, Kathryn A.V. Jones, Giovanna Sifontes, Sandra Valdez, Yiting Zhang, Bernardo A. Mainou.

**Supervision:** Nicholas Wiese, Bernardo A. Mainou.

**Validation:** Nicholas Wiese, Bernardo A. Mainou.

**Visualization:** Nicholas Wiese, Bernardo A. Mainou.

**Writing – original draft:** Nicholas Wiese, Bernardo A. Mainou.

**Writing – review & editing:** Nicholas Wiese, William Hendley, Basit Jafri, Kathryn A.V. Jones, Giovanna Sifontes, Sandra Valdez, Yiting Zhang, Bernardo A. Mainou.

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
