## [Decision Letter · Decision Letter 0]

26 Dec 2025

Dear Dr. Mainou,

Thank you for submitting your manuscript to PLOS ONE. After careful consideration, we feel that it has merit but does not fully meet PLOS ONE’s publication criteria as it currently stands. Therefore, we invite you to submit a revised version of the manuscript that addresses the points raised during the review process.

We look forward to receiving your revised manuscript.

Kind regards,

Maël Bessaud

Academic Editor

PLOS One

Journal Requirements:

Reviewers' comments:

Reviewer's Responses to Questions

**Comments to the Author**

1. Is the manuscript technically sound, and do the data support the conclusions?

Reviewer #1: Partly

Reviewer #2: Yes

2. Has the statistical analysis been performed appropriately and rigorously?

Reviewer #1: Yes

Reviewer #2: Yes

3. Have the authors made all data underlying the findings in their manuscript fully available?

Reviewer #1: Yes

Reviewer #2: Yes

4. Is the manuscript presented in an intelligible fashion and written in standard English?

Reviewer #1: Yes

Reviewer #2: Yes

Reviewer #1: Manuscript Number: PONE-D-25-58771

This paper describes the in house validation of hyper attenuated PV strains (S19) relative to the original Sabin and Salk strains in the poliovirus neutralization assay. The S19 strains are safer and should reduce the use of Salk and Sabin strains in these assays and thereby limit the need for labs to become accredited as poliovirus essential facilities. The paper is clear and quite straight forward: the conclusion is that there is good correlation and the S19 strains can replace the Sabin and Salk strains in standard neutralization assays. Therefore publication is warranted. It is recommended to deal appropriately at least with the 4 generic comments.

Major/generic comments

1/ PV serotypes 1-2-3 are recognized, and IPV vaccination confers good protection against all 3 PV serotypes, and there is supposedly a good correlation between Sabin and Salk strains neutralization titers. - How do you explain the EV-71 high titers pool did react to Salk strains but not to Sabin strains? - How many sera were tested in all assays and can you show data on serotype specific Sabin-Salk titer correlations?

2/ Lines 125-132: It seems none of the assays with the >50 human sera were run in duplicate or triplicate per sample: is that correct? Please explain how the intra assay reproducibility was assessed. If only assessed using the in house ref sera (how many? Add to line 117) this seems to me to be sub-standard. Were samples mentioned in lines 266-277 retested?

3/ An important difference is the temperature: one might expect a statement in the discussion that apparently no relevant effect of the temperature decrease on neutralizing antibody effectivity was noted, as was reported for poliovirus way back (at 27 C) (https://pmc.ncbi.nlm.nih.gov/articles/PMC2137845/pdf/517.pdf)

and for WNV (https://journals.plos.org/plospathogens/article?id=10.1371/journal.ppat.1002111)

4/ Use of S19 for IVIG was already published: Continued use of poliovirus after eradication: hyper‐attenuated strains as a safe alternative for release testing of human immunoglobulins - Farcet - 2018 - Transfusion - Wiley Online Library, this needs to be mentioned in the intro.

Minor/textual comments

Line 19: add “of the USA”

Line 44-45: “humans are the only natural reservoir” is no reason to vaccinate: please rephrase.

Line 52: “areas of poor immunization coverage (12, 13).” Better is to refer to poor hygiene or poor WASH infrastructure, as very good IPV vaccination will still allow multiple transmission cycles of an introduced OPV strain and the rise of VDPVs.

Lines 57-58: BSL3 is not identical to GAPIV, rephrase to GAPIV (throughout the whole text).

Line 58: delete Salk: intended virus culture as in the neutralization assays is not allowed with any WPV without containment (in most countries, I do not know about the USA)

Line 63 and line 100: add UK

Line 74-75: add “individual immunity”as proven SN titers are required for PEF employees, and titers are/can be requested by clinicians in stemcell transplantation procedures.

Line 82: Add reference to other serological assays, including those without the need for infectious virus use altogether: eg : Evaluation of a poliovirus-binding inhibition assay as an alternative to the virus neutralization test - PubMed, A novel multiplex poliovirus binding inhibition assay applicable for large serosurveillance and vaccine studies, without the use of live poliovirus - PubMed; Poliovirus-binding inhibition ELISA based on specific chicken egg yolk antibodies as an alternative to the neutralization test - PubMed; Development of a poliovirus neutralization test with poliovirus pseudovirus for measurement of neutralizing antibody titer in human serum - PubMed.

Line 100: “Individual viruses”, I think you mean “virus strains” and add the MOI or “as per MHRA provided protocol”

Lines 137-147: Move to material & method section?

Lines 148-197 and fig 1 and 2: there is a lot of duplication in txt and figures.

Line 140: abbreviate ‘’polioviruses’’ to PVs, as is done also in previous sentences.

Line 142: same as previous comment. Check manuscript throughout (why not abbreviate in the discussion?).

Line 258-260: This seem to be two independent observations? Please rephrase and/or explain the relevance.

Line 281-282: Rephrase, suggestion: ‘’We suggest that if further passages … , verification of the absence of mutations through sequence analysis as described previously (17) is required.’’

Line 290: Switching from past tense to present tense (‘’we are unable..’’)?

Line 292: This sentence has some grammatical errors. Suggestion to change to: ‘’An additional limitation is the need to modify the standard microneutralization assay when using the S19-PV strains.’’

Reviewer #2: The manuscript by Wiese et al. is sound and long awaited 'field trial' of S19 strains in real lab.

I haven't found any issues, but I have some comments.

Major comment:

Present approach for poliovirus containment according to the GAP IV includes all wild polioviruses, including PV type 1 - they all under containment and require a PEF status of the lab to work with these viruses. Technically, the PEF and non-PEF labs are not BSL-2 and BSL-3 labs; the requirments are very similar, but not the same. Therefore, please, correct throughout the manuscript the lab requirments to work wit different PV, otherwise, it's confusing and does not reflect the present state poliovirus containment.

Minor comments:

L97-98 - what are the numbers in brackets? Cat.No?

L104-106 - please, distinguish the numbers in brackets: cat.No from GenBank IDs

L112 - what is 'PV-positive rat serum..."? Was it a referebnce received from WHO? Please, add some description from the sera passports

L118 - what are residual serum samples? Please, add some desription: human or not, from polio cases or from vaccinees, vaccinated with which vaccine, when collected

L125-132 - was the virus dose controlled as recommended in 27? which doses were accepted/discarded?

L138 - WHO reference sera - which one it is addressed to?

L166 and below - serum titers - please, add log2 or dilution?

L173 - Anti-PV serum - which one it is addressed to?

L198 - In-House Reference Serum (IHRS) - if you've added an abbreviation, please, use it throughout the text

**Do you want your identity to be public for this peer review?** For information about this choice, including consent withdrawal, please see our Privacy Policy

Reviewer #1: No

Reviewer #2: No

You may also use PLOS’s free figure tool, NAAS, to help you prepare publication quality figures: https://journals.plos.org/plosone/s/figures#loc-tools-for-figure-preparation

---

## [Author Response · Author response to Decision Letter 1]

10 Feb 2026

Reviewer comments are addressed below in a point-by-point basis. Additional information is included in 'Response to Reviewers' file.

Reviewer #1:

1/ PV serotypes 1-2-3 are recognized, and IPV vaccination confers good protection against all 3 PV serotypes, and there is supposedly a good correlation between Sabin and Salk strains neutralization titers. - How do you explain the EV-71 high titers pool did react to Salk strains but not to Sabin strains?

The data associated with each pool was tested against Sabin or Salk strains, not both. EV-71 pools were obtained from MHRA and there was not sufficient sample to test the same pool against Sabin and Salk strains. We note the discrepancies in the results with the EV-71 pools in the Discussion (Page 12) and note that differences observed may be due to the EV-71 pools being from human sera. As such, it is possible that the different EV-71 pools obtained could reflect a previous exposure or vaccination to poliovirus by one of the donors in the pool.

How many sera were tested in all assays and can you show data on serotype specific Sabin-Salk titer correlations?

Sera were tested in one independent experiment, with each serum sample tested in triplicate in each testing run with Sabin and Salk parental strains. Sera were tested in three independent experiments, with each serum sample run in triplicate in each testing run with S19-Sabin and S19-Salk strains. Figure legends for each condition indicate how many replicates and independent experiments were performed for each condition tested. We assessed levels of neutralizing antibody levels between Sabin and Salk strains and did not observe significant differences between both strains (see figure below which is not in manuscript) and text added on Page 10.

2/ Lines 125-132: It seems none of the assays with the >50 human sera were run in duplicate or triplicate per sample: is that correct?

For experiments testing human sera with Sabin (Figure 4) and Salk (Figure 5) strains, each serum sample was tested in one independent experiment, with each serum sample tested in triplicate, for parental Sabin and Salk and three independent experiments with each sample tested in triplicate for S19-Sabin and S19-Salk strains. The number of independent experiments performed were limited by available sample volume for each condition.

Please explain how the intra assay reproducibility was assessed. If only assessed using the in house ref sera (how many? Add to line 117) this seems to me to be sub-standard. Were samples mentioned in lines 266-277 retested?

The intra-assay reproducibility was calculated using in-house reference sera with known neutralizing antibody levels against parental Sabin and Salk strains. Nine individual in-house reference serum samples, each tested in triplicate, were tested across three independent experiments against the Sabin-S19 strains (Figure 3A). Four individual in-house reference serum samples, each tested in triplicate, were tested across three independent experiments against Salk-S19 strains (Figure 4).

3/ An important difference is the temperature: one might expect a statement in the discussion that apparently no relevant effect of the temperature decrease on neutralizing antibody effectivity was noted, as was reported for poliovirus way back (at 27 C) (https://pmc.ncbi.nlm.nih.gov/articles/PMC2137845/pdf/517.pdf)

and for WNV (https://journals.plos.org/plospathogens/article?id=10.1371/journal.ppat.1002111)

We have added language to the Discussion on Page 16 to highlight that temperature does not impact overall neutralizing antibody levels detected.

4/ Use of S19 for IVIG was already published: Continued use of poliovirus after eradication: hyper‐attenuated strains as a safe alternative for release testing of human immunoglobulins - Farcet - 2018 - Transfusion - Wiley Online Library, this needs to be mentioned in the intro.

This reference was added (see reference number 18) to support text in Intro on Page 4.

Minor/textual comments

All minor/textual comments have been addressed in the manuscript with track changes:

Line 19: add “of the USA”

Line 44-45: “humans are the only natural reservoir” is no reason to vaccinate: please rephrase.

Line 52: “areas of poor immunization coverage (12, 13).” Better is to refer to poor hygiene or poor WASH infrastructure, as very good IPV vaccination will still allow multiple transmission cycles of an introduced OPV strain and the rise of VDPVs.

Lines 57-58: BSL3 is not identical to GAPIV, rephrase to GAPIV (throughout the whole text).

Line 58: delete Salk: intended virus culture as in the neutralization assays is not allowed with any WPV without containment (in most countries, I do not know about the USA)

Line 63 and line 100: add UK

Line 74-75: add “individual immunity”as proven SN titers are required for PEF employees, and titers are/can be requested by clinicians in stemcell transplantation procedures.

Line 82: Add reference to other serological assays, including those without the need for infectious virus use altogether: eg : Evaluation of a poliovirus-binding inhibition assay as an alternative to the virus neutralization test - PubMed, A novel multiplex poliovirus binding inhibition assay applicable for large serosurveillance and vaccine studies, without the use of live poliovirus - PubMed; Poliovirus-binding inhibition ELISA based on specific chicken egg yolk antibodies as an alternative to the neutralization test - PubMed; Development of a poliovirus neutralization test with poliovirus pseudovirus for measurement of neutralizing antibody titer in human serum - PubMed.

Line 100: “Individual viruses”, I think you mean “virus strains” and add the MOI or “as per MHRA provided protocol”

Lines 137-147: Move to material & method section?

Lines 148-197 and fig 1 and 2: there is a lot of duplication in txt and figures.

Line 140: abbreviate ‘’polioviruses’’ to PVs, as is done also in previous sentences.

Line 142: same as previous comment. Check manuscript throughout (why not abbreviate in the discussion?).

Line 258-260: This seem to be two independent observations? Please rephrase and/or explain the relevance.

Line 281-282: Rephrase, suggestion: ‘’We suggest that if further passages … , verification of the absence of mutations through sequence analysis as described previously (17) is required.’’

Line 290: Switching from past tense to present tense (‘’we are unable..’’)?

Line 292: This sentence has some grammatical errors. Suggestion to change to: ‘’An additional limitation is the need to modify the standard microneutralization assay when using the S19-PV strains.’’

Reviewer #2: The manuscript by Wiese et al. is sound and long awaited 'field trial' of S19 strains in real lab.

I haven't found any issues, but I have some comments.

Major comment:

Present approach for poliovirus containment according to the GAP IV includes all wild polioviruses, including PV type 1 - they all under containment and require a PEF status of the lab to work with these viruses. Technically, the PEF and non-PEF labs are not BSL-2 and BSL-3 labs; the requirements are very similar, but not the same. Therefore, please, correct throughout the manuscript the lab requirements to work with different PV, otherwise, it's confusing and does not reflect the present state poliovirus containment.

Due to changes in GAP IV implementation in the United States by the U.S. National Authority for Containment of Poliovirus (NAC), we addressed containment requirements to those stated by the US NAC. For domestic work with containable materials, engineering requirements by the NAC dictate that PEFs operate in BSL-3 lab spaces. Text has been revised to reflect NAC policies regarding poliovirus containable materials.

Minor comments:

All minor/textual comments have been addressed in the manuscript with track changes.

L97-98 - what are the numbers in brackets? Cat.No?

L104-106 - please, distinguish the numbers in brackets: cat.No from GenBank IDs

L112 - what is 'PV-positive rat serum..."? Was it a referebnce received from WHO? Please, add some description from the sera passports

L118 - what are residual serum samples? Please, add some desription: human or not, from polio cases or from vaccinees, vaccinated with which vaccine, when collected

L125-132 - was the virus dose controlled as recommended in 27? which doses were accepted/discarded?

L138 - WHO reference sera - which one it is addressed to?

L166 and below - serum titers - please, add log2 or dilution?

L173 - Anti-PV serum - which one it is addressed to?

L198 - In-House Reference Serum (IHRS) - if you've added an abbreviation, please, use it throughout the text

---

## [Editor Report · Decision Letter 1]

11 Feb 2026

Evaluation of Hyper-Attenuated S19 Poliovirus Strains for Use in Poliovirus Neutralization Assays

PONE-D-25-58771R1

Dear Dr. Mainou,

We’re pleased to inform you that your manuscript has been judged scientifically suitable for publication and will be formally accepted for publication once it meets all outstanding technical requirements.

Kind regards,

Maël Bessaud

Academic Editor

PLOS One
---

## [Editor Report · Acceptance letter]

PONE-D-25-58771R1

PLOS One

Dear Dr. Mainou,

I'm pleased to inform you that your manuscript has been deemed suitable for publication in PLOS One. Congratulations! Your manuscript is now being handed over to our production team.

Kind regards,

on behalf of

Dr. Maël Bessaud

Academic Editor

PLOS One